# About Universality and Thermodynamics of Turbulence

**DOI:** 10.3390/e21030326

**Published:** 2019-03-26

**Authors:** Damien Geneste, Hugues Faller, Florian Nguyen, Vishwanath Shukla, Jean-Philippe Laval, Francois Daviaud, Ewe-Wei Saw, Bérengère Dubrulle

**Affiliations:** 1SPEC, CEA, CNRS, Université Paris-Saclay, CEA Saclay, 91191 Gif-sur-Yvette, France; 2CNRS, ONERA, Arts et Metiers ParisTech, University of Lille, Centrale Lille, FRE 2017-LMFL-Laboratoire de Mécanique des Fluides de Lille—Kampé de Fériet, F-59000 Lille, France; 3School of Atmospheric Sciences, Sun Yat-sen University, Guangzhou 510275, China

**Keywords:** turbulence, intermittency, multifractal, thermodynamics

## Abstract

This paper investigates the universality of the Eulerian velocity structure functions using velocity fields obtained from the stereoscopic particle image velocimetry (SPIV) technique in experiments and direct numerical simulations (DNS) of the Navier-Stokes equations. It shows that the numerical and experimental velocity structure functions up to order 9 follow a log-universality (Castaing et al. *Phys. D Nonlinear Phenom.* 1993); this leads to a collapse on a universal curve, when units including a logarithmic dependence on the Reynolds number are used. This paper then investigates the meaning and consequences of such log-universality, and shows that it is connected with the properties of a “multifractal free energy”, based on an analogy between multifractal and thermodynamics. It shows that in such a framework, the existence of a fluctuating dissipation scale is associated with a phase transition describing the relaminarisation of rough velocity fields with different Hölder exponents. Such a phase transition has been already observed using the Lagrangian velocity structure functions, but was so far believed to be out of reach for the Eulerian data.

## 1. Introduction

A well-known feature of any turbulent flow is the Kolmogorov-Richardson cascade by which energy is transferred from large to small length scales until the Kolmogorov length scale below which it is removed by viscous dissipation. This energy cascade is a non-linear and an out-of-equilibrium universal process. Moreover, the corresponding non-dimensional energy spectrum $E\left(k\right)/\epsilon^{2/3}\eta^{5/3}$ is an universal function of kη, where $\eta ={\left(\nu^{3}/\epsilon \right)}^{1/4}$ is the Kolmogorov length scale, ϵ the mean energy dissipation rate per unit mass, and ν the kinematic viscosity. Every used quantity is identified with its definition in a nomenclature available in Table 1. However, there seems to be little dependences on the Reynolds number, boundary, isotropy or homogeneity conditions [1]. In facts, the energy spectrum is based upon a quantity, the velocity correlation that is quadratic in velocity. Nevertheless, it is now well admitted that the universality does not carry over for statistical quantities that involve higher order moments. For example, the velocity structure functions of order *p*, given by $S_{p}\left(\ell \right)={\langle ∥u\left(x+r\right)-u\left(x\right)∥}^{p}\rangle_{x,∥r∥=\ell }$ are not universal, at least when expressed in units of the Komogorov scale η and velocity $u_{K}={\left(\nu \epsilon \right)}^{1/4}$ (see below, Section 3.2 for an illustration).

The mechanism behind this universality breakage is identified in [3], where a generalization of the Kolmogorov theory is introduced, based on the hypothesis that a turbulent flow is multifractal. In this model, the velocity field is locally characterized by a Hölder exponent *h*, such that $\delta_{\ell }u\left(x\right)\equiv \langle ∥u\left(x+r\right)-{u\left(x\right)∥\rangle }_{∥r∥=\ell }∼\ell^{h(x)}$; here *h* is a stochastic function that follows a large deviation property [4] $P\left(\log(|\delta_{\ell }u\left|/u_{0}\right)=h\log\left(\ell /L_{0}\right)\right)∼{\left(\ell /L_{0}\right)}^{C(h)}$, where $u_{0}$ (resp. $L_{0}$) is the characteristic integral velocity (resp. length), and C(h) is the multifractal spectrum. Velocity fields with h<1 are rough in the limit ℓ→0. Indeed they are at least not differentiable. In real flows, any rough field with h>−1 can be regularized at sufficiently small scale (the “viscous scale”) by viscosity. The first computation of such dissipative scale was performed by Paladin and Vulpiani [5], who showed that it scales with viscosity like $\eta_{h}\propto \nu^{1/(1+h)}$, thereby generalizing the Kolmogorov scale, which corresponds to h=1/3. Such a dissipative scale fluctuates in space and time (along with *h*), resulting in non-universality for high order moments, at least when expressed in units of η and $u_{K}$.

A few years later, Frisch and Vergassola [6] claimed that the universality of the energy spectrum can be recovered, if the fluctuations of the dissipative length scale are taken into account by introducing a new non-dimensionalisation procedure. The new prediction was that logE(k)ϵ−23η−53/log(Re) should be a universal function of log(kη)/log(Re), where Re is the Reynolds number. This claim was examined by Gagne et al., later using data from the Modane wind tunnel experiments [7]. They further suggested that the prediction can be extended to the velocity structure functions $S_{p}$, so that log(Sp(ℓ)/uKp)/log(Re) should be a universal function of log(ℓ/η)/log(Re), at any given *p*. They found good agreement for *p* up to 6. The velocity measurements, in the above experiments, were performed using hot wire anemometry, which provide access to only one component of velocity. To our knowledge, no further attempts have been made to check the claim with more detailed measurements.

The purpose of the present paper is to reexamine this claim. However, now using the velocity fields obtained from the Stereoscopic Particle Image Velocimetry (SPIV) in experiments and the direct numerical simulations (DNS) of the Navier-Stokes equations (NSE). We show that the numerical and experimental velocity structure functions up to order 9 follow a log-universality [7]; they indeed collapse on a universal curve, if we use units that include log(Re) dependence. We then investigate the meaning and consequences of such a log-universality, and show that it is connected with the properties of a “multifractal free energy”, based on an analogy between multifractal and thermodynamics (see [8] for summary). This framework uses co-existing velocity fields with different Hölder exponents which are regularized at variable scales. We show that in such a framework, this fluctuating dissipation length scale is associated with a phase transition describing the relaminarisation of velocity fields.

## 2. Experimental and Numerical Setup

### 2.1. Experimental Facilities and Parameters

We use experimental velocity field described in [9]. The radial, axial and azimuthal velocity are measured in a Von Kármán flow, using Stereoscopic Particle Image Velocimetry technique at different resolutions Δx. The Von Kármán flow is generated in a cylindrical tank of radius R=10 cm through counter-rotation of two independent impellers with curved blades. The flow was maintained in a turbulent state at high Reynolds number by two independent impellers, rotating at various frequencies. Figure 1 shows the sketch of the experimental setup. The five experiments are performed in conditions so that the non-dimensional mean energy dissipation per unit mass is constant. The viscosity is monitored using mixture of water and glycerol, so as to vary the Kolmogorov length η. Table 2 summarizes the different parameters; $R_{\lambda }=\lambda u^{\mathrm{rms}}/\nu$ is the Reynolds number based on the Taylor length scale $\lambda =\sqrt{\frac{\langle u^{2}\rangle }{\langle \nabla u^{2}\rangle }}$, the root mean squared velocity $u^{\mathrm{rms}}$ and the kinematic viscosity ν.

All velocity measurements are performed in a vertical plane that contains the rotation axis. The case (A) corresponds to measurements over the whole plane contained in between the two impellers, and extending from one side to the other side of the cylinder. Its resolution is 5 to 10 times coarser than similar measurements performed by zooming on a region centered around the symmetry point of the experiment (on the rotation axis, half way in between the two impellers), over a square window of size 4 cm×3 cm. Since the flow is not homogeneous, statistics in this central region may differ from statistics computed over the whole tank. This explains the strong difference of $R_{\lambda }$ between (A) and (B,C). The little differences between (B) and (C) are explained by the different experimental resolutions used.

### 2.2. Direct Numerical Simulation

The direct numerical simulations (DNS), based on pseudo-spectral methods, are performed in order to compare with our experimental data. The DNS runs with $R_{\lambda }=25$, $R_{\lambda }=80$, $R_{\lambda }=90$ and $R_{\lambda }=138$ are performed using the NSE solver VIKSHOBHA [10], whereas the run with $R_{\lambda }=53$ is carried out using another independent pseudo-spectral NSE solver. The velocity field u is computed on a 2π triply-periodic box.

Turbulent flow in a statistically steady state is obtained by using the Taylor-Green type external forcing in the NSE at wavenumber $k_{f}=1$ and amplitude $f_{0}=0.12$, the value of viscosity is varied in order to obtain different values of $R_{\lambda }$ (see Ref. [10] for more details).

## 3. Theoretical Background

### 3.1. Velocity Increments vs. Wavelet Transform (WT) of Velocity Gradients

The classical theories of Kolmogorov [11,12] are based on the scaling properties of the velocity increment, defined as $\delta_{\ell }u\left(x,t\right)=\langle ∥u\left(x+r,t\right)-{u\left(x,t\right)∥\rangle }_{∥r∥=\ell }$ where ℓ=∥r∥ is the distance over which the increment is taken. As pointed out by [8], a more natural tool to characterize the local scaling properties of the velocity field is the wavelet transform of the tensor $\partial_{j}u_{i}$, defined as:
(1)$$G_{ij}\left(x,\ell ,t\right)=\int \nabla_{j}\Phi_{\ell }\left(r\right)u_{i}\left(x+r,t\right)dr$$
where $\Phi_{\ell }\left(x\right)=\ell^{-3}\Phi \left(x/\ell \right)$ is a smooth function, non-negative with unit integral. In what follows, we choose a Gaussian function $\Phi \left(x\right)={\exp(-∥x∥}^{2}{/2)/\left(2\pi \right)}^{\frac{3}{2}}$ such that ∫Φ(r)dr=1. We then compute the wavelet velocity increments as
(2)$$\delta W\left(x,\ell ,t\right)=\ell \max_{ij}\left|G_{ij}\left(x,\ell ,t\right)\right|$$

This formulation is especially well suited for the analysis of the experimental velocity field, as it naturally allows to average out the noise. It has been verified that the wavelet-based approach yields the same values for the scaling exponents as those computed from the velocity increments [10].

### 3.2. K41 and K62 Universality

In the first theory of Kolmogorov [11], the turbulence properties depend only on two parameters: the mean energy dissipation per unit mass ϵ and the viscosity ν. The only velocity and length unit that one can build using these quantities are the Kolmogorov length $\eta ={\left(\nu^{3}/\epsilon \right)}^{1/4}$ and velocity $u_{K}={\left(\epsilon \nu \right)}^{1/4}$. The structure functions are then self-similar in the inertial range $\eta \ll \ell \ll L_{0}$, where $L_{0}$ is the integral scale, and follow the universal scaling:
(3)$$S_{p}\left(\ell \right)\equiv \langle {\left(\delta_{\ell }u\right)}^{p}\rangle ∼u_{K}^{p}{\left(\frac{\ell }{\eta }\right)}^{p/3}$$
which can also be recast into:
(4)$${\tilde{S}}_{p}\left(\ell \right)\equiv \frac{S_{p}\left(\ell \right)}{{\left(S_{3}\left(\ell \right)\right)}^{p/3}}=C_{p}$$
where $C_{p}$ is a (non universal) constant.

This scaling is typical of a global scale symmetry solution, and was criticized by Landau, who considered it incompatible with observed large fluctuations of the local energy dissipation. Kolmogorov then built a second theory (K62), in which fluctuations of energy dissipation were assumed to follow a log-normal statistics, and taken into account via an intermittency exponent κ and a new length scale *L*, thereby breaking the global scale invariance. The resulting velocity structure functions then follow the new scaling:
(5)$$S_{p}\left(\ell \right)∼{\left(\epsilon \ell \right)}^{p/3}{\left(\frac{\ell }{L}\right)}^{\kappa p(3-p)}$$
which implies a new kind of universality involving the relative structure functions ${\tilde{S}}_{p}$ as:(6)$${\tilde{S}}_{p}\left(\ell \right)\equiv \frac{S_{p}\left(\ell \right)}{{\left(S_{3}\left(\ell \right)\right)}^{p/3}}∼A_{p}{\left(\frac{\ell }{L}\right)}^{\tau (p)}$$
where τ(p)=κp(3−p) and $A_{p}$ is a constant. Such a formulation already predicts an interesting universality, if $L=L_{0}$, as we should have:
(7)$${\left(\frac{L_{0}}{\eta }\right)}^{\tau (p)}{\tilde{S}}_{p}\left(\ell \right)∼A_{p}{\left(\frac{\ell }{\eta }\right)}^{\tau (p)}$$

Therefore, we should be able to collapse all structure functions, at different Reynolds number by plotting ${\left(\frac{L_{0}}{\eta }\right)}^{\tau (p)}{\tilde{S}}_{p}$ as a function of $\frac{\ell }{\eta }$, given that L0/η∼Re3/4. There is however no clear prediction about the value of *L* and we show in the data analysis (Section 4) that *L* differs from $L_{0}$.

The relation (Equation 7) shows that $\log\left({\left(\frac{L_{0}}{\eta }\right)}^{\tau (p)}{\tilde{S}}_{p}\right)$ is a linear function of $\log(\frac{\ell }{\eta })$. In principle, such universal scaling is not valid outside the inertial range, i.e., for example when ℓ<η. To be more general than previously thought, it can however be shown using the multifractal formalism as first shown by [6].

### 3.3. Multifractal and Fluctuating Dissipation Length

For the multifractal (MFR) model, it is assumed that the turbulence is locally self-similar, so that there exists a scalar field h(x,ℓ,t), such that
(8)$$h\left(x,t,\ell \right)=\frac{\log\left(\delta_{\ell }u\left(x,t\right)/u_{0}\right)}{\log(\ell /L)}$$
for a range of scales in a suitable “inertial range” $\eta_{h}\ll \ell \ll L$, where *L* is a large inertial scale, $\eta_{h}$ a cut-off length scale, and $u_{0}$ a characteristic large-scale velocity. This scale $\eta_{h}$ is a generalization of the Kolmogorov scale, and is defined as the scale where the local Reynolds number $\ell |\delta_{\ell }u|/\nu$ is equal to 1. Writing $\delta_{\ell }u=u_{0}{\left(\ell /L\right)}^{h}$ leads to the expression of $\eta_{h}$ as a function of the global Reynolds number Re=u0L/ν as ηh∼LRe−1/(1+h). This scale thus appears as a fluctuating cut-off which depends on the scaling exponent and therefore on x. This is the generalization of the Kolmogorov scale η∼LRe−3/4≡η13, and was first proposed in [5]. Below $\eta_{h}$, the velocity field becomes laminar, and $\delta_{\ell }u\propto \ell$. When the velocity field is turbulent, $h\equiv \log\left(\delta_{\ell }u/u_{0}\right)/\log\left(\ell /L\right)$ varies stochastically as a function of space and time. Also, if the turbulence is statistically homogeneous, stationary and isotropic, *h* only depends on *ℓ*, the scale magnitude. Therefore, formally, *h* can be regarded as a continuous stochastic process labeled by log(ℓ/L). By Kramer’s theorem [13], one sees that as in the limit ℓ→0, log(L/ℓ)→∞, we have:
(9)$$P\left[\log\left(\delta_{\ell }u/u_{0}\right)=h\log\left(\ell /L\right)\right]∼e^{\log(\ell /L)C(h)}={\left(\frac{\ell }{L}\right)}^{C(h)}$$
where C(h) is the rate function of *h*, also called multifractal spectrum. Formally, C(h) can be interpreted as the co-dimension of the set where the local Hölder exponent at scale *ℓ* is equal to *h*. Using Gärtner-Elis theorem [13], one can connect *C* and the velocity structure functions as:(10)$$S_{p}\left(\ell \right)=\langle {\left(\delta_{\ell }u\right)}^{p}\rangle =\int_{h_{\min}}^{h_{\max}}u_{0}^{p}{\left(\frac{\ell }{L}\right)}^{ph+C(h)}dh$$

To proceed further and make connection with previous section, we set $\epsilon =u_{0}^{3}/L$ so that $S_{p}\left(\ell \right)$ can now be written:
(11)$$S_{p}\left(\ell \right)={\left(\epsilon \ell \right)}^{p/3}\int_{h_{\min}}^{h_{\max}}{\left(\frac{\ell }{L}\right)}^{p(h-1/3)+C(h)}dh∼{\left(\epsilon \ell \right)}^{p/3}{\left(\frac{\ell }{L}\right)}^{\tau (p)}$$

This shows that τ(p) is the Legendre transform of the rate function C(h+1/3), i.e., $\tau \left(p\right)=\min_{h}\left(p\left(h-1/3\right)+C\left(h\right)\right)$, and equivalently, that C(h) is the Legendre transform of τ(p). Because of this, it is necessarily convex. The set of points where C(h)≤3, represents the set of admissible or observable *h*, is therefore necessarily an interval, bounded by $-1\leq h_{\min}$ and $h_{\max}\leq 1$.

As noted by [6], the scaling exponent ζ(p)=p/3+τ(p) defined via Equation (Equation 11) is only constant in a range of scale where $\ell >\eta_{h}$ for any $h\in [h_{\min},h_{\max}]$. For small enough *ℓ*, this condition is not met anymore, since as soon as $\ell <\eta_{h}$, all velocity fields corresponding to *h* are “regularized”, and do not contribute anymore to intermittency since they scale like *ℓ*. This results in a slow dependence of ζ(p) with respect to the scale, which is obtained via the corrected formula:
(12)$$S_{p}\left(\ell \right)={\left(\epsilon \ell \right)}^{p/3}\int_{\eta_{h}\leq \ell }{\left(\frac{\ell }{L}\right)}^{p(h-1/3)+C(h)}dh∼{\left(\epsilon \ell \right)}^{p/3}{\left(\frac{\ell }{L}\right)}^{\tau (p,\ell )}$$

To understand the nature of the correction, we can compute the value of *h* such that $\ell =\eta_{h}$. This gives: h(ℓ)=−1+log(Re)/log(L/ηh). With θ=log(L/ℓ)/log(Re), Equation (Equation 12) can be rewritten as:(13)S˜p(ℓ)≡Sp(ℓ)S3(ℓ)p/3=∫−1+1/θhmaxℓLp(h−1/3)+C(h)dh∼exp−θτ(p,θ)log(Re)
where τ(p,θ)=τ(p) when $\theta \leq 1/(1+h_{\max})$ and τ(p,θ)=p(θ−1/3)+C(−1+1/θ) when $1/\left(1+h_{\max}\right)\leq \theta \leq 1/\left(1+h_{\min}\right)$. As discussed by [6], this implies a new form of universality that extends beyond the inertial range, into the so-called extended dissipative range, as;
(14)log(S˜p)log(Re)=−τ(p,θ)θ,θ=log(L/ℓ)/log(Re)

If the scale *L* is constant and equal to $L_{0}$, the integral scale, then we have Re=(L0/η)4/3 and the multifractal universality implies that $\log\left({\tilde{S}}_{p}\right)/\log\left(L_{0}/\eta \right)$ is a function of $\log\left(\ell /\eta \right)/\log\left(L_{0}/\eta \right)$. When the function is linear, we thus recover the K62 universality. The multifractal universality is thus a *generalization* of the K62 universality.

This form of universality is however not easy to test, as the scale *L* is not known a priori, and may still depend on Re. In what follows, we demonstrate a new form of universality that allows more freedom upon *L* and encompass both K62 and multifractal universality.

### 3.4. General Universality

Using the hypothesis that turbulence maximizes some energy transfer in the scale space, Castaing [2] suggested a new form of universality for the structure functions, that reads:(15)γ(Re)logSp(ℓ)ApuKp=Gp,γ(Re)log(ℓK0/η)
where $A_{p}$ and $K_{0}$ are universal constants and β and *G* are general functions, *G* being linear in the inertial range, G(p,x)∼τ(p)x. The validity of this universal scaling was checked by Gagne and Castaing [7] on data obtained from the velocity fields measured in a jet using hot wire anemometry. They found good collapse of the structure functions at different Taylor Reynolds $R_{\lambda }$, provided γ(Re) is constant at low Reynolds numbers and follows a law of the type: γ(Re)∼γ0/log(Rλ/R*), where $R_{*}$ is a constant, whenever $R_{\lambda }>400$. Since we have Rλ∼Re1/2 and (L0/η)∼Re3/4, we can rewrite Equation (Equation 15) as:(16)β(Re)log(S˜p(ℓ)/S0p)log(L0/η)=Hp,β(Re)log(ℓ/η)log(L0/η)
where $S_{0p}$ are some constants and β and *H* are general functions. Compared to the K62 or MFR universality Formulas (Equation 7) or (Equation 14), we see that Formula (Equation 16) is a generalization of these two universality with $L=L_{0}$. It allows however more flexibility than K62 or MFR universality through the function β(Re), which is a new fitting function. We test these predictions in Section 4 and provide a physical interpretation of (Equation 16) in Section 5.

## 4. Check of Universality Using Data Analysis

The various universality are tested using the velocity structure functions based on the wavelet velocity increments Equation (Equation 2), in order to minimize the noise in the experimental data. We define:(17)$$S_{p}\left(\ell \right)={\langle |\delta W\left(x,\ell ,t\right)|}^{p}\rangle_{x,t}$$

We then apply this formula to both experimental data (Table 2) and numerical data (Table 3), to get wavelet velocity structure functions at various scales and Reynolds numbers.

### 4.1. Check of K41 Universality

The K41 universality (Equation 3) can be checked by plotting:(18)$$\log\left(\frac{S_{p}}{u_{K}^{p}}\right)=F\left(\log\left(\frac{\ell }{\eta }\right)\right)$$

This is shown in Figure 2 for both experimental and numerical data. Obviously, the data do not collapse on a universal curve, meaning that K41 universality does not hold. This is well known, and is connected to intermittency effects [14].

### 4.2. Check of K62 Universality

The K62 universality (Equation 7) can be checked by plotting:(19)$$\log\left[{\left(\frac{L_{0}}{\eta }\right)}^{\tau (p)}{\tilde{S}}_{p}\right]=F\left(\log\left(\frac{\ell }{\eta }\right)\right)$$

The collapse depends directly on τ(p), the intermittency exponents. Obtaining the best collapse of all curves is in fact a way to fit the best scaling exponents τ(p). We thus implement a minimization algorithm that provides the values of τ(p) that minimized the distance between the curve and the line of slope τ(p). The values of τ(p) are reported in Table 4. The best collapse is shown on Figure 3a for the DNS, and Figure 3b for the experiment. The collapse is better for experiments than for the DNS. However, in both cases, there are significant differences in between points at different $R_{\lambda }$, at larger scales, showing that universality is not yet reached.

### 4.3. Check of General Universality

We can now check the most general universality, by plotting:(20)β(Re)log(S˜p/S0p)log(L0/η)=Hp,β(Re)log(ℓ/η)log(L0/η)

In this case, best collapse is obtained by fitting two families of parameters: $S_{0p}$, β(Re) that are obtained through a procedure of minimization. We take the DNS at $R_{\lambda }=138$ as the reference case, and find for both DNS and experiments, the values of β(Re) and $S_{0p}$ that best collapse the curves. The corresponding collapses are provided in Figure 5. The collapses are good for any value of Re, except for the DNS at the lowest Reynolds number, which does not collapse in the far dissipative range.

### 4.4. Function β(Re)

Motivated by earlier findings by [7], we plot in Figure 6 the value 1/β as a function of $R_{\lambda }$.

Our results are compatible with $1/\beta ∼\beta_{0}/\log\left(R_{\lambda }\right)$, with $\beta_{0}∼4/3$ over the whole range of Reynolds number. For comparison, we provide also on Figure 6 the values found by Gagne and Castaing [7] in jet of liquid Helium, shifted by an arbitrary factor to make our values coincide with them at large Reynolds number. This shift is motivated by the fact that β(Re) is determined up to a constant, depending upon the amplitude of the structure functions used as reference. At large Reynolds, our values are compatible with theirs. At low Reynolds, however, we do not observe the saturation of 1/β that is observed in the jet experiment of [7]. An interpretation of the meaning of β(Re) is provided in Section 5.

### 4.5. Scaling Exponents

Our Collapse method enables us to obtain the scaling exponents of the structure functions ζ(p) by the following two methods:

(i) Using the K62 universality, we get τ(p), and then ζ(p)=ζ(3)p/3+τ(p). These estimates still depend on the value of ζ(3), which is not provided by the K62 universality plot. To obtain them, we use a minimization procedure on both experimental $\log(S_{3}/u_{K}^{3})$ from the one hand, and the numerical $\log(S_{3}/u_{K}^{3})$ on the other hand (see Figure 4a), to compute ζ(3) as the value that minimizes the distance between the curve and a straight line of slope ζ(3). The values so obtained are reported in Table 4, and are used to compute ζ(p) from τ(p). In Table 4, two different methods are used to process the experimental data. The subscript SAW refers to the values obtained by [9] on the same set of experimental data, using velocity increments and Extended Self-Similarity technique [15]. The quantities with subscript EXP are computed through a least square algorithm upon τ(p), minimizing the scatter of the rescaled structure functions $\log\left[{\left(\frac{L_{0}}{\eta }\right)}^{\tau (p)}{\tilde{S}}_{p}\right]$ with respect to the line ${\left(\ell /\eta \right)}^{\tau (p)}$. DNS data have been processed the same way as EXP.

(ii) Using the general universality, we may also get $\tau_{p,\mathrm{univ}}$ by a linear regression on the collapse curve. Please note that since the data are collapsed, this provides a very good estimates of this quantity, with the lowest possible noise. In practice, we observe no significant differences with the two estimates; therefore, we only report the values obtained by following the first method.

The corresponding values are plotted in Figure 4 and summarized in Table 4. Please note that for both DNS and experiments, the value of ζ(3) is different from 1, which is apparently incompatible with the famous Kolmogorov 4/5th law that predicts ζ(3)=1. This is because we use *absolute* values of wavelet increments, while the Kolmogorov 4/5th law uses signed values. We have checked that using unsigned values, we obtain a scaling that is closer to 1, but with larger noise. Note also that when we consider the relative value ζ(p)/ζ(3), we obtain values that are close to the values obtained [9] on the same set of experimental data, using velocity increments and Extended Self-Similarity technique [15].

### 4.6. Multifractal Spectrum

From the values of ζ(p), one can get the multifractal spectrum C(h) by performing the inverse Legendre transform:
(21)$$C\left(h\right)=\min_{p}\left[ph+\zeta \left(p\right)\right]$$

Practically, this allows to use the following formula:
(22)$$C\left(\frac{d\;\zeta (p)}{d\;p}{|}_{p^{*}}\right)=\zeta \left(p^{*}\right)-p^{*}\frac{d\;\zeta (p)}{d\;p}{|}_{p^{*}}$$

To estimate *C*, we thus first perform a polynomial interpolation of order 4 on ζ(p), then derivate the polynomial to estimate $\frac{d\;\zeta (p)}{d\;p}$, thus get *C* through Equation (Equation 22). The result is provided in Figure 6b for both the DNS and the experiment.

The curve looks like a portion of parabola, corresponding to a log-normal statistics for the wavelet velocity increments. Specifically, fitting by the shape:
(23)$$C\left(h\right)=\frac{{\left(h-a\right)}^{2}}{2b}$$
we get a=0.35 and b=0.045. This parabola also provides a good fit of the scaling exponents, as shown in Figure 4 by performing Legendre transform of C(h) given by Equation (Equation 23).

## 5. Thermodynamics and Turbulence

### 5.1. Thermodynamical Analogy

Multifractal obeys a well-known thermodynamical analogy [8,16,17] that will be useful to interpret and extend the general universality unraveled in the previous section. Indeed, considering the quantity:(24)$$\mu_{\ell }=\frac{|\delta W_{\ell }{|}^{3}}{\langle |\delta W_{\ell }{|}^{3}\rangle }$$

By definition $\mu_{\ell }$ is positive definite and $\langle \mu_{\ell }\rangle =1$ for any *ℓ*. It therefore can be interprated as a scale dependent measure. It then also follows a large-deviation property as:
(25)$$P\left[\log\left(\mu_{\ell }\right)=E\log\left(\ell /\eta \right)\right]∼e^{\log(\ell /\eta )S(E)}$$
where S(E) is the large deviation function of $\log(\mu_{\ell })$ and has the meaning of an energy while log(ℓ/η) has the meaning of a volume, and $\log\left(\mu_{\ell }\right)/\log\left(\ell /\eta \right)$ is an energy density. With the definition of $\mu_{\ell }$, it is easy to see that *S* is connected to *C*, the large deviation function of $|\delta W_{\ell }|$. In fact, since in the inertial range where $\langle |\delta W_{\ell }{|}^{3}\rangle ∼\ell^{\zeta (3)}$, we have S(E)=C(3h−ζ(3)). By definition, we also have:
(26)$${\tilde{S}}_{3p}=\frac{S_{3p}}{S_{3}^{p}}=\langle e^{p\log(\mu_{\ell })}\rangle$$
so that ${\tilde{S}}_{3p}$ is the partition function associated with the variable $\log(\mu_{\ell })$, at the pseudo-inverse temperature $p=1/k_{B}T$. Taking the logarithm of the partition function ${\tilde{S}}_{3p}$, we then get the free energy *F* as:
(27)$$F=\log({\tilde{S}}_{3p})$$

By the Gärtner-Elis theorem, *F* is the Legendre transform of *S*: $F=\min_{E}\left(pE-S\left(E\right)\right)$. The free energy a priori depends on the temperature $T=1/k_{B}p$, on the volume V=log(ℓ/η) and on the number of degrees of freedom of the system *N*. If we identify N=(1/β(Re))log(L0/η), we see that the general universality means:(28)$$F\left(T,V,N\right)=NF\left(T,\frac{V}{N},1\right)$$
i.e., can be interpreted as *extensivity* of the free energy.

The thermodynamic analogy is thus meaningful and is summarized in Table 5. It can be used to derive interesting prospects.

### 5.2. Multifractal Pressure and Phase Transition

Given our free energy, $F=\log({\tilde{S}}_{3p})$, we can also compute the quantity conjugate to the volume, i.e., the multifractal pressure as: P=∂F/∂V. In the inertial range, where ${\tilde{S}}_{p}∼\ell^{\tau (p)}$, we thus get P=τ(p), which only depends on the temperature. Outside the inertial range, *P* has the meaning of a local scaling exponent that also depends upon the scale, i.e., on the volume *V* and on *N* (Reynolds number). Using our universal functions derived in Figure 5, we can then compute empirically the multifractal pressure *P* and see how it varies as a function of *T*, *V* and *N*. It is provided in Figure 7 for $R_{\lambda }=25$ and $R_{\lambda }=53$, and in Figure 8 for $R_{\lambda }=90$ and $R_{\lambda }=138$. We see that at low Reynolds number, the pressure decreases monotonically from the dissipative range, reaches a lowest points and then increases towards the largest scale. There is no clear flat plateau that would correspond to an “inertial” range. In contrast, at higher Reynolds number, a plateau appears for p=1 to p=4 when going towards the largest scale, the value of the plateau corresponding to $\tau_{\mathrm{DNS}}$. The plateau transforms into an inflection point for p≥5 making the derivative ∂P/∂V change sign. This is reminiscent of a phase transition occurring in the inertial range, with coexistence of two phases: one “laminar” and one “turbulent”. We interpret such a phase transition as the result of the coexistence of region of flows with different Hölder exponents, with areas where the flow has been regularized due to the action of viscosity, because of the random character of the dissipative scale (see below).

## 6. Conclusions

We show that a deep analogy exists between multifractal and classical thermodynamics. In this framework, one can derive from the usual velocity structure function an effective free energy that respects the classical extensivity properties, provided one uses several degrees of freedom (given by N=1/β(Re)) that scales like $\log(R_{\lambda }$). This number is much smaller than the classical N∼Re9/4 that is associated with the number of nodes needed to discretize the Navier-Stokes equation down to the Kolmogorov scale. It would be interesting to see whether this number is also associated with the dimension of a suitable “attractor of turbulence”. Using the analogy, we also find the “multifractal” equation of state of turbulence, by computing the multifractal “free energy” *F* and “pressure” P=∂F/∂V. We find that for large enough $R_{\lambda }$ and *p* (the temperature), the system obeys a phase transition, with coexistence of phase like in the vapor-liquid transition. We interpret this phase transition as the result of the coexistence of region of flows with different Hölder exponents, with areas where the flow is relaminarized due to the action of viscosity, because of the random character of the dissipative scale. We note that this kind of phenomenon has already been observed in the context of Lagrangian velocity increments, using the local scaling exponent $\zeta \left(p,\Delta t\right)=d\left(\log\left(S_{p}\left(\Delta t\right)\right)\right)/d\left(\log\left(\Delta t\right)\right)$ [18]. The phase transition is then associated with the existence of a fluctuating dissipative time scale. It is further shown that in a multifractal without fluctuating dissipative time scale, the local exponent decreases monotonically from dissipative scale to large scale, implying a disappearance of the phase transition [19]. 

## Figures and Tables

**Figure 1 entropy-21-00326-f001:**
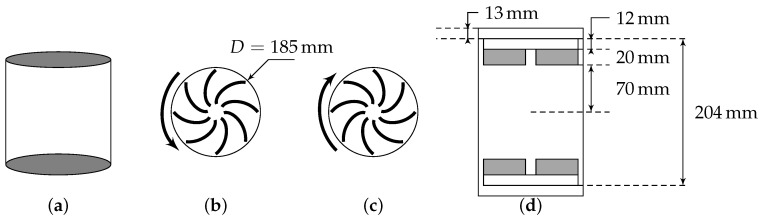
Von Kármán swirling flow generator. (**a**) normal view, bottom (**b**) and top (**c**) impellers rotating -both seen from the center of the cylinder, and (**d**) sketch with the relevant measures. A device not shown here maintains the temperature constant during the experiment. Both impellers are counter-rotating.

**Figure 2 entropy-21-00326-f002:**
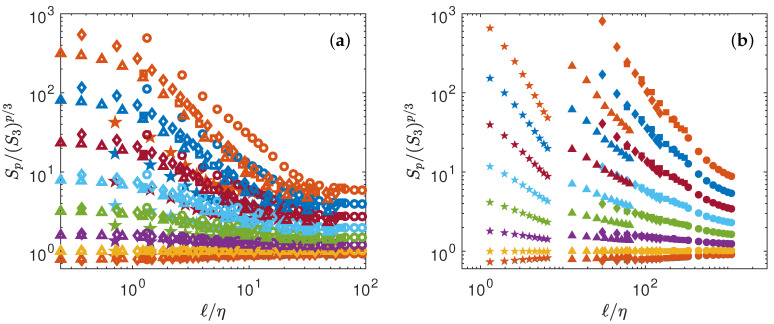
Test of K41 universality Equation (Equation 4). (**a**) Numerical data (**b**) Experimental data. The structure functions have been shifted by arbitrary factors for clarity and are coded by color: p=1: blue symbols; p=2: orange symbols; p=3: yellow symbols; p=4: magenta symbols; p=5: green symbols; p=6: light blue symbols; p=7: red symbols; p=8: blue symbols; p=9: orange symbols. For K41 universality to hold, all the function should be constant, for a given *p*.

**Figure 3 entropy-21-00326-f003:**
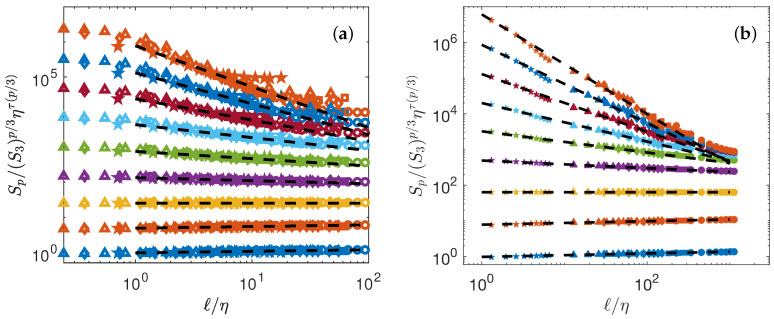
Test of K62 universality Equation (Equation 7). (**a**) Numerical data (**b**) Experimental data. The structure functions are shifted by arbitrary factors for clarity and are coded by color: p=1: blue symbols; p=2: orange symbols; p=3: yellow symbols; p=4: magenta symbols; p=5: green symbols; p=6: light blue symbols; p=7: red symbols; p=8: blue symbols; p=9: orange symbols. The dashed lines are power laws with exponents τ(p)=ζ(p)−ζ(3)p/3, with ζ(p) shown in Figure 4a.

**Figure 4 entropy-21-00326-f004:**
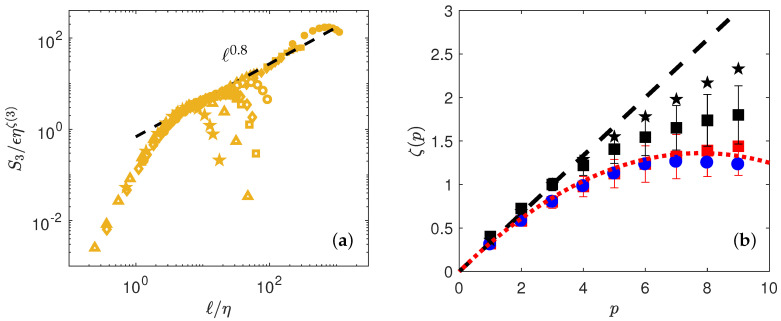
(**a**) Determination of ζ(3) by best collapse using both DNS (open symbols) and experiments (filled symbols). The black dashed line is $\ell^{0.8}$. (**b**) Scaling exponents ζ(p) of the wavelet structure functions of δW as a function of the order, from Table 4, for DNS (blue circle) and experiments (red square). The red dotted line is the function $\min_{h}\left(hp+C\left(h\right)\right)$ with C(h) given by $C\left(h\right)={\left(h-a\right)}^{2}/2b$, with a=0.35 and b=0.045. The black stars correspond to $\zeta_{\mathrm{SAW}}\left(p\right)/\zeta_{\mathrm{SAW}}\left(3\right)$ (see Table 4), while the black squares correspond to $\zeta_{\mathrm{EXP}}\left(p\right)/\zeta_{\mathrm{EXP}}\left(3\right)$.

**Figure 5 entropy-21-00326-f005:**
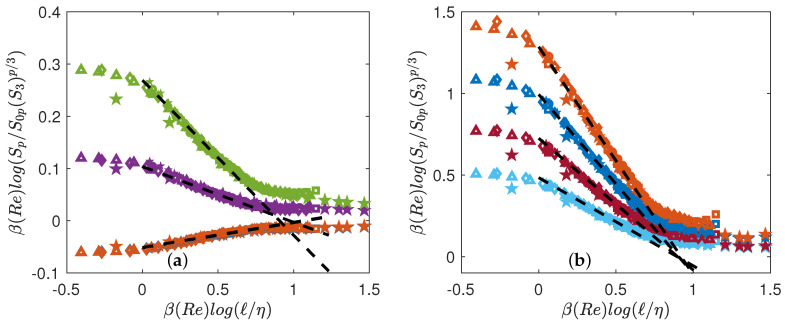
Test of general universality Equation (Equation 20) using both DNS (open symbols) and experiments (filled symbols). The functions are coded by color. (**a**) p=1: blue symbols; p=2: orange symbols; p=4: magenta symbols; p=5: green symbols; (**b**) p=6: light blue symbols; p=7: red symbols; p=8: blue symbols; p=9: orange symbols. The functions have been shifted by arbitrary factors for clarity. The dashed lines are power laws with exponents τ(p)=ζ(p)−ζ(3)p/3, with ζ(p) shown in Figure 4a.

**Figure 6 entropy-21-00326-f006:**
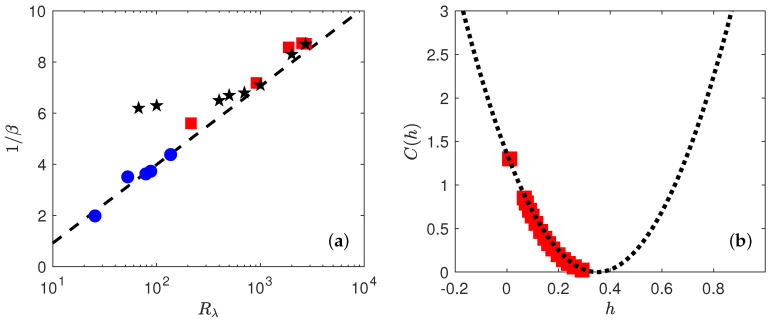
(**a**) Variation of 1/β(Re) versus $\log(R_{\lambda })$ in experiments (red square) and DNS (blue circle) when using the DNS at $R_{\lambda }=138$ as the reference case. Black stars correspond to the values found by Gagne and Castaing in [7] shifted by an arbitrary factor to coincide the values at large Reynolds. The black dashed line is $\left(4/3\right)\log\left(R_{\lambda }/5\right)$. (**b**) Multifractal spectrum C(h) for the experiments. The spectrum is obtained by taking inverse Legendre transform of the scaling exponents ζ(p) shown in Figure 4. The dotted line is a parabolic fit $C\left(h\right)={\left(h-a\right)}^{2}/2b$ with a=0.35 and b=0.045.

**Figure 7 entropy-21-00326-f007:**
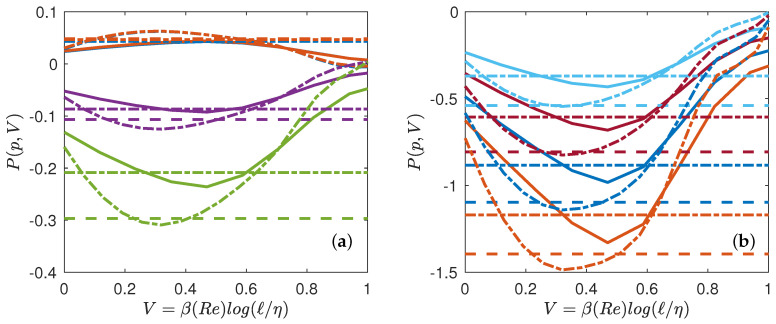
Multifractal equation of state of turbulence. Multifractal pressure as a function of the volume for $\;R_{\lambda }=25$ (line), $\;R_{\lambda }=53$ (dashed-dotted line). The functions are coded by color. (**a**) p=1: blue symbols; p=2: orange symbols; p=4: magenta symbols; p=5: green symbols; (**b**) p=6: light blue symbols; p=7: red symbols; p=8: blue symbols; p=9: orange symbols. The colored dotted line (resp. dashed dotted line) are values corresponding to $P\left(p,V\right)=\tau_{\mathrm{EXP}}\left(p\right)$ (resp. $\tau_{\mathrm{DNS}}\left(p\right)$), that are reported in Table 4.

**Figure 8 entropy-21-00326-f008:**
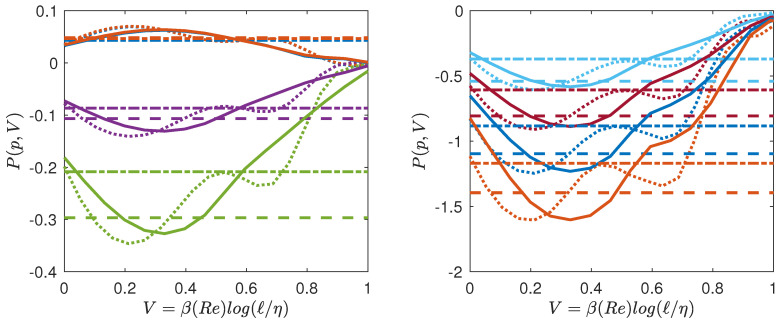
Same as Figure 7 for $\;R_{\lambda }=90$ (line), $\;R_{\lambda }=138$ (dotted line). Note the inflexion point appearing in the curves.

**Table 1 entropy-21-00326-t001:** Nomenclature.

Symbol	Mathematical Definition	Interpretation
u(x,t)	$\in R^{3}\times R\to R^{3}$	Velocity field
*k*	$\in R_{+}$	Wavenumber
E(k)	FT$\left({\langle u_{i}\left(x+r,t\right)u_{i}\left(x,t\right)\rangle }_{x,∥r∥=\ell ,t}\right)$	Energy spectrum
$k_{f}$	$\in R_{+}^{*}$	Forcing wavenumber
$N_{x}$	∈N	Grid size in direction *x*
ν	$\in R_{+}^{*}$	Kinematic viscosity
ϵ	$\in R_{+}^{*}$	Mean dissipation power per unit mass
η	${\left(\frac{\nu^{3}}{\epsilon }\right)}^{\frac{1}{4}}$	Kolmogorov scale
$u_{K}$	${\left(\nu \epsilon \right)}^{\frac{1}{4}}$	Kolmogorov velocity
$u_{0}$	$\in R_{+}^{*}$	Characteristic velocity
$L_{0}$	$\in R_{+}^{*}$	Characteristic length
Re	$\frac{u_{0}L_{0}}{\nu }$	Reynolds number
λ	$\sqrt{\frac{{\langle u^{2}\rangle }_{x,t}}{{\langle \nabla u^{2}\rangle }_{x,t}}}$	Taylor length
$u^{\mathrm{rms}}$	$\sqrt{{\langle u^{2}\rangle }_{x,t}-{\langle u\rangle }_{x,t}^{2}}$	Root mean squared velocity
$R_{\lambda }$	$\frac{\lambda u^{\mathrm{rms}}}{\nu }$	Taylor Reynolds number
Δx	$\in R_{+}^{*}$	SPIV spatial resolution
*p*	∈[1,9]	Power
*ℓ*	$\in R_{+}^{*}$	Scale
*L*	$\in R_{+}^{*}$	Inertial large scale
$\delta_{\ell }u\left(x,t\right)$	$\langle ∥u\left(x+r,t\right)-{u\left(x,t\right)∥\rangle }_{∥r∥=\ell }$	Velocity increment at scale *ℓ*
Φ(x)	${\exp(-∥x∥}^{2}{/2)/\left(2\pi \right)}^{\frac{3}{2}}$	Wavelet filter
$\Phi_{\ell }\left(x\right)$	$\ell^{-3}\Phi \left(x/\ell \right)$	Wavelet filter at scale *ℓ*
$G_{ij}\left(x,\ell ,t\right)$	$\int \nabla_{j}\Phi_{\ell }\left(r\right)u_{i}\left(x+r,t\right)dr$	Wavelet transform of ∇u
δW(x,ℓ,t)	$\ell \max_{ij}\left|G_{ij}\left(x,\ell ,t\right)\right|.$	Wavelet velocity increment
$S_{p}\left(\ell \right)$	$\left\{\begin{array}{cc} {\langle {\left(\delta_{\ell }u\right)}^{p}\rangle }_{x,t} & \mathrm{In}\;\mathrm{theory}\\ {\langle {\left(\delta W\left(x,\ell ,t\right)\right)}^{p}\rangle }_{x,t} & \mathrm{For}\;\mathrm{data}\;\mathrm{analysis} \end{array}\right.$	Velocity structure function
${\tilde{S}}_{p}\left(\ell \right)$	$\frac{S_{p}}{S_{3}^{p/3}}$	Relative structure function
h(x,t)	$\in R^{3}\times R\to \left[-1,1\right]$	Local Hölder exponent
C(h)	$P\left(\log(|\delta_{\ell }u\left|/u_{0}\right)=h\log\left(\ell /L_{0}\right)\right)∼{\left(\ell /L_{0}\right)}^{C(h)}$	Multifractal Spectrum
$\eta_{h}$	L0Re−11+h	Multifractal regularization scale
κ	$\in R_{+}^{*}$	Intermittency parameter
τ(p)	κp(3−p)	Lognormal Intermittency correction
ζ(p)	$\frac{p}{3}+\tau \left(p\right)$	Scaling exponent
θ(ℓ)	log(L/ℓ)log(Re)	Rescaled length
τ(p,θ)	$\left\{\begin{array}{ccc} \tau (p) & \mathrm{if} & \theta \leq \frac{1}{1+h_{\max}}\\ p\left(\theta -\frac{1}{3}\right)+C\left(-1+\frac{1}{\theta }\right) & \mathrm{if} & \frac{1}{1+h_{\max}}\leq \theta \leq \frac{1}{1+h_{\min}} \end{array}\right.$	General intermittency correction
τ(p,ℓ)	τ(p,θ(ℓ))	General intermittency correction
γ(Re),β(Re)	$R_{+}\to R$	Fitting functions
*G*	$R^{2}\to R$	General function from Castaing [2]
$A_{p}$, $K_{0}$	γ(Re)logSpApuKp=Gp,γ(Re)log(ℓK0/η)	Universal parameters
*H*	$R^{2}\to R$	New general function
$S_{0p}$	β(Re)log(S˜p/S0p)log(L0/η)=Hp,β(Re)log(ℓ/η)log(L0/η)	Universal parameter
*a*, *b*	$C\left(h\right)=\frac{{\left(h-a\right)}^{2}}{2b}$	Parabolic fit
$\beta_{0}$	$1/\beta \left(R_{\lambda }\right)∼\beta_{0}/\log\left(R_{\lambda }\right)$	Parameter
$\tau_{p,\mathrm{univ}}$	$\frac{\tau (p,\ell )}{\log(\ell /L)}$ for *ℓ* in Inertial range	Intermittency correction from general rescaling
$\mu_{\ell }\left(x\right)$	$\frac{\delta W{\left(x,\ell \right)}^{3}}{<\delta W{\left(y,\ell \right)}^{3}{>}_{y}}$	Spatial scale dependent measure
S(E)	$P\left[\log\left(\mu_{\ell }\right)=E\log\left(\ell /\eta \right)\right]∼e^{\log(\ell /\eta )S(E)}$	Large deviation function of $\log(\mu_{\ell })$
$k_{B}$	$\in R_{+}^{*}$	Boltzmann constant
*T*	$1/k_{B}p$	Temperature
*E*	$\log(\mu_{\ell })$	Energy
*N*	log(Re)	Number of degrees of freedom
*V*	log(ℓ/η)	Volume
*P*	τ(p,ℓ)	Pressure
*F*	$\log({\tilde{S}}_{3p})$	Free energy

**Table 2 entropy-21-00326-t002:** Parameters for the 5 experiments realized (A,B,C,D and E). F is the rotation frequency of the discs, Re refers to the Reynolds number based on the diameter of the tank, $R_{\lambda }$ is the Reynolds based on the Taylor micro-scale. η gives the estimated Kolmogorov length according to the experiment and Δx refers to the spatial resolution of SPIV measurements. The second last column gives the number of frames over which are calculated the statistics. Except for (E), the Reynolds are much larger than those available with DNS. Table adapted from [10].

Case	Frequency (Hz)	Glycerol Part	Re	$R_{\lambda }$	η (mm)	Δx	Frames	Symbol
A	5	0%	$3\times 10^{5}$	$1.9\times 10^{3}$	0.02	2.4	$3\times 10^{4}$	○
B	5	0%	$3\times 10^{5}$	$2.7\times 10^{3}$	0.02	0.48	$3\times 10^{4}$	□
C	5	0%	$3\times 10^{5}$	$2.5\times 10^{3}$	0.02	0.24	$2\times 10^{4}$	◊
D	1	0%	$4\times 10^{4}$	$9.2\times 10^{2}$	0.08	0.48	$1\times 10^{4}$	△
E	1.2	59%	$6\times 10^{3}$	$2.1\times 10^{2}$	0.37	0.24	$3\times 10^{4}$	⋆

**Table 3 entropy-21-00326-t003:** Parameters for the DNS. $R_{\lambda }$ is the Reynolds based on the Taylor micro-scale. η is the Kolmogorov length. The third column gives resolution of the simulation through $k_{\max}\eta$, where $k_{\max}=N_{x}/3$ is the maximum wavenumber. The fourth column gives the grid size; notice that the dimensionless length of the box is 2π. Here, $\ell_{\min}$ is the smallest scale available for the calculations of the wavelets. $k_{f}$ is the forcing wavenumber. The Sample columns gives the number of points (frames × grid size) over which the statistics are computed.

$R_{\lambda }$	η	$k_{\max}\eta$	$N_{x}\;\times N_{y}\;\times \;N_{z}$	$\ell_{\min}/\eta$	Samples	Symbol
25	0.079	3.35	$128^{3}$	0.635	5000	⋆
53	0.034	8.5	$768^{3}$	0.31	105,000	△
80	0.020	1.68	$256^{3}$	1.22	270,000	□
90	0.017	5.7	$1024^{3}$	0.36	10,000	◊
138	0.009	1.55	$512^{3}$	1.37	12,000	○

**Table 4 entropy-21-00326-t004:** Scaling exponents τ(p) and ζ(p) found by the collapse method based on K62 universality for experimental data (subscript EXP) or numerical data (subscript DNS). The subscript SAW refers to the values obtained by [9]. The exponents $\tau_{\mathrm{EXP}}\left(p\right)$ (red square) and $\tau_{\mathrm{DNS}}$ (blue circle) have been computed through a least square algorithm.

Exponent\Order	p=1	p=2	p=3	p=4	p=5	p=6	p=7	p=8	p=9
$\zeta_{\mathrm{SAW}}/\zeta_{\mathrm{SAW}}\left(3\right)$	0.36	0.69	1	1.29	1.55	1.78	1.98	2.17	2.33
$\zeta_{\mathrm{DNS}}$	0.31	0.58	0.80	0.98	1.12	1.23	1.26	1.25	1.23
$\zeta_{\mathrm{EXP}}$	0.32	0.58	0.80	0.98	1.12	1.23	1.32	1.39	1.44
$\tau_{\mathrm{DNS}}$	0.04	0.05	0	−0.09	−0.21	−0.37	−0.61	−0.88	−1.17
$\tau_{\mathrm{EXP}}$	0.05	0.05	0	−0.09	−0.21	−0.36	−0.54	−0.74	−0.96

**Table 5 entropy-21-00326-t005:** Summary of the analogy between the multifractal formalism of turbulence and thermodynamics.

	Thermodynamics	Turbulence
Temperature	$k_{B}T$	1/p
Energy	*E*	$\log(\mu_{\ell })$
Number of d.f.	*N*	log(Re)=log(L0/η)/β0
Volume	*V*	log(ℓ/η)
Pressure	*P*	τ(p,ℓ)
Free energy	*F*	$\log({\tilde{S}}_{3p})$

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
