# Peer review of "About Universality and Thermodynamics of Turbulence"

_entropy, 2019, doi:10.3390/e21030326_

Round 1

Reviewer 1 Report

This paper investigates turbulence from a thermodynamic point of view. Experiments and detailed numerical simulations are performed to analyse one turbulent flow field and data is analysed to formulate an analogy between thermodynamic and turbulent phenomena. This is an interesting paper and involves rigorous analyses. There are some grammatical and spelling errors. Also, some variables are not defined; a nomenclature could be added.

Author Response

We thank the referree for his constructive remarks.
We have followed the referree’s suggestions and have added a nomenclature and corrected many  grammatical and spelling errors.

Reviewer 2 Report

This paper focuses on the Eulerian velocity structure functions from both experimental and numerical data in steady turbulence. Structure functions of up to 9th order were investigated, with a particular focus on the universality of these structure functions with the logarithm of the separation variable r and the Reynolds number. A phase transition is identified and is related to a fluctuating dissipation scale.

In my opinion, the paper is very well written with a good balance between details and to-the-point discussions of the results. There is a clear objective and conclusion. The English is good, and I recommend publication of this paper in ‘entropy’. Please forward my minor comments to the authors for implementation.

1.    I believe the paper could have an higher impact when the discussion focuses a bit more on the physics. For instance, when making statements such as “…fluctuating dissipation length scale is associated with a phase transition describing the relaminarisation of rough velocity fields with different Holder exponents.”, most readers won’t follow this. Rough velocity fields? Different Holder exponents of what? Is a relaminarisation intermittent?

2.    Typos appear throughout the manuscript. Please proofread extensively. E.g. abstract: “themodynamics”, page 2: “caracteristic”, etc.

3.    Without having to go to reference [8], the paper benefits from a bit more discussion of the experimental cases. Why have case A, B and C different Re_lambda? Etc.

Author Response

We thank the referree for his constructive remarks.

1) A few sentences have been rewritten so as to focus more on the physics.

2) we have corrected many grammatical and spelling errors

3) More explanations have been added regarding the experimental conditons and the values of different R_lambda.

Reviewer 3 Report

This paper investigates the universality of a velocity structure function and shows an analogy between multifractal and classical thermodynamics. However, the title is really rather too broad, although it is reasonable. In other words, it is also hard for reviewer to judge “Universality and Thermodynamics of Turbulence”. While for some minor comments about the paper written, some comments are as follows:

1.    The Abstract narrates in the first-person and it would be better to use the third-person perspective.

2.    Pay more attention to the tense.

3.    Recheck the whole passage to make sure there are no more misspelled words, for instance, ‘caracteristic’ in page 2, line 3.

4.    There are too many words below figures and tables and it would be better to make some adjustments to these necessary explanations. 

5.    Please add a part of Nomenclature to define the physical quantities occur in the paper, especially those in tables and equations. And it would increase the readability for readers.

6.    The grammar and sentence construction throughout the paper need to be improved. Conjunctions, singular and plural forms used in the whole passage shall be better checked.

7.    Why does the illustration part of Figure 4 appear after that of Figure 6-b? Please alter the order of figures.

8.    It would be better to reorganize the Abstract and Conclusion part to increase the logicality, which can help readers to find the research points much easier.

Author Response

We thank the referree for his constructive remarks. Below is a point by point answer to his suggestion.

0)We have slightly modified the title by adding ‘About’ so as to restrict the generality of the title.

1)We have followed the referree’s suggestion and rewritten the abstract using the third person.

2) The tense has been standardized.

3) We have corrected many grammatical and spelling errors

4)Parts of the captions have been re-integrated in the main text.

5)A nomenclature has been added.

6) We have corrected many grammatical and spelling errors

7)Figures have been reorder in in a more logical sequence.

8)We have partially rewritten abstract and conclusion.